# The Role of Health Information Sources on Cervical Cancer Literacy, Knowledge, Attitudes and Screening Practices in Sub-Saharan African Women: A Systematic Review

**DOI:** 10.3390/ijerph21070872

**Published:** 2024-07-03

**Authors:** Joyline Chepkorir, Dominique Guillaume, Jennifer Lee, Brenice Duroseau, Zhixin Xia, Susan Wyche, Jean Anderson, Hae-Ra Han

**Affiliations:** 1Institute of Clinical and Translational Research, Johns Hopkins University, Baltimore, MD 21202, USA; 2School of Nursing, Johns Hopkins University, Baltimore, MD 21205, USA; dguilla2@jhmi.edu (D.G.); hhan3@jhu.edu (H.-R.H.); 3Jhpiego, a Johns Hopkins University Affiliate, Baltimore, MD 21231, USA; 4International Vaccine Access Center, International Health Department, Johns Hopkins Bloomberg School of Public Health, Johns Hopkins University, Baltimore, MD 21205, USA; 5Department of Pediatrics, Johns Hopkins University School of Medicine, Baltimore, MD 21205, USA; jlee694@jh.edu; 6Department of Hematological Malignancies, Johns Hopkins Hospital, Baltimore, MD 21287, USA; zxia14@jhmi.edu; 7Department of Media and Information, Michigan State University, East Lansing, MI 48824, USA; 8Department of Gynecology and Obstetrics, Johns Hopkins School of Medicine, Johns Hopkins University, Baltimore, MD 21287, USA; 9Department of Health, Behavior, and Society, Johns Hopkins Bloomberg School of Public Health, Johns Hopkins University, Baltimore, MD 21205, USA

**Keywords:** cervical cancer screening, cervical cancer knowledge, cervical cancer literacy, cancer screening attitudes, health information sources, cancer prevention

## Abstract

Cervical cancer is the leading cause of cancer deaths among Sub-Saharan African women. This systematic review aimed to identify information sources and their relation to cervical cancer knowledge, literacy, screening, and attitudes. Peer-reviewed literature was searched on 2 March 2022, and updated on 24 January 2023, in four databases—CINAHL Plus, Embase, PubMed, and Web of Science. Eligible studies included those that were empirical, published after 2002, included rural women, and reported on information sources and preferences. The quality of the selected articles was assessed using the Mixed Methods Appraisal Tool. Data extraction was conducted on an Excel spreadsheet, and a narrative synthesis was used to summarize findings from 33 studies. Healthcare workers were the most cited information sources, followed by mass media, social networks, print media, churches, community leaders, the Internet, and teachers. Community leaders were preferred, while healthcare workers were the most credible sources among rural women. There was generally low cervical cancer knowledge, literacy, and screening uptake, yet high prevalence of negative attitudes toward cervical cancer and its screening; these outcomes were worse in rural areas. A content analysis revealed a positive association of health information sources with cervical cancer literacy, knowledge, screening, and positive screening attitudes. Disparities in cervical cancer prevention exist between rural and urban Sub-Saharan African women.

## 1. Introduction

Cervical cancer (CC) is the leading cause of cancer-related deaths among Sub-Saharan African women, mainly between the ages 21 and 48 years [1]. In fact, Sub-Saharan Africa (SSA) has the highest rates of CC globally, owing to limited HPV vaccination, and sub-optimal cervical cancer screening (CCS) uptake among sexually active women [1]. Only about 14% of women in SSA between ages 30 and 49 have ever been screened for CC in their lifetime and approximately 12% have been screened at least twice by age 45, which is lower than the 70% target stipulated by the World Health Organization [2]. It is estimated that 60 to 80% of women diagnosed with CC in SSA are rural residents, who tend to seek medical attention when CC has advanced. Only about 0.4 to 14% of rural-based Sub-Saharan African women are screened for pre-cancerous lesions compared to at least 20% of urban residents [3]. The lower screening uptake among rural women has been attributed to limited healthcare resources and infrastructure in most parts of rural SSA [4]. This challenge is compounded by poor health literacy, misinformation, and lack of information on CC [5,6,7,8]. 

CC information has been established as a critical correlate of CCS uptake in diverse populations [9,10]. Research on Sub-Saharan African women has consistently revealed suboptimal CCS uptake rates, and limited CC knowledge [11,12] and CC literacy [5,13] as well as negative attitudes towards CCS [11,12]. Particularly, basic information on the causes of CC, its risk factors and signs and symptoms, the benefits of screening in early detection of CC, and the location of screening services, is sparsely disseminated to women in SS Africa [14]. Some of the negative attitudes that tend to hinder CCS include fear of the screening procedure and fatalistic beliefs about CC [15]. Nonetheless, the role of information sources in shaping women’s CC knowledge, literacy, and attitudes toward screening and screening uptake is rarely explored. Research is also limited in characterizing information sources considered acceptable and credible in the context of cervical cancer prevention among women in SSA. To design interventions aimed at preventing CC and reducing CC burden attributable to limited access to reliable and accurate CC information [13], it is essential to understand the role of health information sources in CC knowledge, literacy, and attitudes toward screening and screening uptake among Sub-Saharan African women, especially in rural areas where screening uptake rates are very low.

This systematic review aimed to, (1) identify the sources of CC-related information accessed and used by Sub-Saharan African women, (2) understand the perceived credibility and preferences for CC information sources, and (3) understand the relationships among health information sources of CCS uptake, CC knowledge, literacy, and attitudes toward screening. CC screening literacy was operationally defined as the degree to which women can obtain, process, understand, and communicate information related to cervical cancer screening; while CC knowledge was defined as the vocabulary and conceptual understanding of CC information [16].

## 2. Methods

### 2.1. Inclusion and Exclusion Criteria

Studies were eligible for inclusion if they: (a) involved women residing in rural SSA, (b) reported on CC-related information sources, (c) mentioned credibility of or preferences for specific sources of CC information, (d) were empirical studies, (e) were written in English, and (f) were published after 2002. Records were excluded if they were: (a) greay literature, (b) abstracts with no full texts, (c) about HPV vaccination only, (d) publications on CC treatment only, (e) conducted among urban and/or sub-urban populations only, (f) did not indicate the geographical location of the study (rural/urban/sub-urban), (g) publications on CC educational interventions with no mention of other CC information sources, and (h) were about other cancer types or overall health. 

### 2.2. Information Sources and Search Strategy

A literature search was conducted on 2 March 2022, and updated on 24 January 2023, to determine whether there was any newly published evidence on the research topic. A medical librarian assisted in searching for key terms in four electronic databases—CINAHL Plus, Embase, PubMed, and Web of Science. The key search terms were cervical neoplasm*, cervical cancer, health information, literacy, and Africa South of the Sahara. We used advanced search functions in each database including “MeSH terms”, searching in “All fields” and connecting the components of each search string using Boolean operators “AND”, and “OR”. Appendix A shows the detailed search strategies applied.

### 2.3. Study Screening and Selection

The records searched were imported to Covidence software (Covidence, Melbourne, Australia), a web-based software designed to streamline systematic and non-systematic literature reviews [17]. Duplicates (*n* = 1756) were automatically removed by Covidence. The remaining articles were screened in two stages. Firstly, a title and abstract review was performed to exclude irrelevant articles. Secondly, a full-text review was conducted to select articles that met the inclusion criteria. Each record was screened by two independent reviewers, with discrepancies resolved through consensus in regular meetings.

### 2.4. Data Extraction Process

The data were extracted by two individuals using an Excel spreadsheet. The extracted data were the first author and publication year, study title, study design, study setting (country), geographical setting (rural/suburban/urban), study population (demographics and sample size), CC information sources, credibility and preferences, CCS practices, attitudes toward CCS, and CC knowledge, association of information sources with CCS uptake, knowledge, literacy, and screening attitudes. 

### 2.5. Quality Assessment

Each study was assessed for quality by two independent reviewers using the Mixed Methods Appraisal Tool, which is intended to assess the design, conduct, and analysis of qualitative, quantitative descriptive, and mixed methods studies quality [18]. Any discrepancies between the reviewers were discussed and reconciled through consensus.

### 2.6. Data Synthesis

Due to clinical and methodological heterogeneity, a narrative synthesis, as opposed to a meta-analysis, was used to report the findings of this review [19]. This approach allowed the identification and analysis of major themes from the extracted data, enabling the authors to summarize the findings [19].

## 3. Results

### 3.1. Study Selection

Based on the selection process, 171 full-text articles were excluded, yielding 33 articles eligible for inclusion. Figure 1 below shows the preferred reporting items for systematic reviews and meta-analysis (PRISMA) flow diagram of the records identified, screened, and included [20].

### 3.2. Features of Included Studies

Table 1 shows a summary of the results.

Among the studies included, there were five qualitative [3,21,22,23,24], twenty-four descriptive cross-sectional [11,14,15,25,26,27,28,29,30,31,32,33,34,35,36,37,38,39,40,41,42,43,44,45], two analytic cross-sectional [46,47], one unmatched case control [48], and one concurrent mixed-methods study [13]. The studies were conducted in 13 out of 48 countries in SSA, with most studies (*n* = 11) conducted in Ethiopia [15,28,30,33,35,40,41,43,44,45,48], followed by Tanzania (*n* = 4) [26,34,37,47]; Uganda (*n* = 4) [13,31,36,38]; Nigeria (*n* = 3) [14,29,39]; two each from Kenya [11,21] and South Africa [27,46]; and one each from Burkina Faso [25], Ghana [23], Rwanda [3], Malawi [24], Namibia [32], Zambia [22], and Zimbabwe [42].

Sixteen studies were either conducted in rural areas [13,14,21,23,24,29,47] or in predominantly rural areas [3,11,15,26,33,35,36,38,41]; twelve studies were in predominantly urban and semi-urban populations [25,30,31,32,39,40,42,43,44,45,46,48]. Four studies were conducted among rural and urban women but the proportions of participants from each residential area were unspecified [22,28,34,37]. About half of the studies (*n* = 16) recruited participants from community settings [11,13,15,21,23,24,26,28,29,31,33,36,38,39,41,42] and 38% (*n* = 13) recruited from health facilities [3,14,22,25,27,30,34,37,43,44,45,46,48]. Other recruitment sites were educational institutions [35,40], both the community and health facility settings [47], and nationally (with unspecified recruitment strategy) [32].

The proportion of rural women in studies that sampled from predominantly rural sites was between 53.3% and 90.3% [3,11,15,26,33,35,36,38,41], while in predominantly urban and semi-urban samples, the proportion of rural women was between 2.3% and 45.6% [3,11,15,25,26,30,31,32,33,35,36,38,39,40,41,42,43,44,45,46,48]. The age range of the study participants was between 15 and 85 years. Among the studies not conducted in educational institutions and in which literacy levels were reported, 4% to 34% of participants were illiterate [11,13,14,15,25,26,28,30,32,33,34,36,39,41,43,44,45,47,48], except for two studies with predominantly rural women in Ethiopia, in which 64% and 41% of the participants were illiterate [15,41].

### 3.3. Quality Appraisal

A summary of quality assessment of included literacy is shown on Table 2 below. Most studies met all five methodological quality criteria used to assess each, based on the study design [3,11,14,15,21,22,23,24,25,26,28,30,31,32,33,35,36,37,38,39,40,41,42,44,45,46,47,48]. Only one study failed to meet two of five quality assessment criteria due to failure to specify the sampling strategy and ambiguity in its exclusion and inclusion criteria [27]. Therefore, reviewers were unable to determine whether the sampling strategy was relevant in addressing the research question and if the sample was representative. One study did not report the number of surveys distributed and it was impossible to establish the existence of non-response bias [29]. One mixed methods study failed to report on the divergences and inconsistencies between qualitative and quantitative results [13]. Two studies neither pre-tested adapted instruments nor reported their reliability, hence reviewers could not establish the appropriateness of the measures in addressing the research questions [34,43].

### 3.4. Sources of Information on CC and CCS

Among participants who had received any form of information about CC and CCS, healthcare workers (i.e., nurses, doctors, community health workers) were the most frequently stated sources [3,11,13,14,15,22,23,24,26,27,28,29,30,31,32,33,35,36,37,38,39,40,41,42,43,44,45,46,48], followed by mass media, such as, radio, television, and newspapers [3,13,26,27,28,30,32,33,35,36,40,42,44]; social networks, such as friends, family/relatives, and neighbors [11,27,29,30,37,44]; print media including brochures, newspapers, magazines, books, and posters [28,29,32,33,40]; and schools [22,23,26,29]. Less frequently reported sources included religious leaders [30,40], churches [26,42], village leaders [21,24], the Internet [14], teachers, religious fellowships, and girls’ clubs [23]. Another innovative community-based government initiative for CC information dissemination was the Women’s Development Army used in Ethiopia [48].

Social media (e.g., WhatsApp, Facebook, and Instagram) was reported as a source of CC information in two studies [11,23]. Specifically, social media was the second (after health care providers) most endorsed information source on CC and CCS in a Ghanaian rural sample that consisted “mostly of a young population” of women aged 19–29 years [23]. In another study involving a predominantly rural sample of women (*n* = 451) aged 18–85 years in Kenya, it was the least endorsed (6% of women) [11].

Studies that included both rural and urban residents showed some differences in CC information sources. For example, while healthcare workers were a predominant source of information in both rural and urban residents among women in a study in Zambia, urban residents had more diverse sources of CC-related information from reliable institutions, including workplaces and schools, compared to rural residents, who mostly received CC information from their social circles, including friends, family, and community outreach programs [22]. Among Tanzanian women (*n* = 575), receiving information from churches and radios was more common in rural residents, and television was more common in urban residents [26].

Only one qualitative study among rural Kenyan women reported on preferences for CC-related information [21]. Researchers found that regardless of their gender, community leaders and village leaders were viewed by “most study participants” as essential channels for CC information dissemination because of their unique understanding of effective strategies to communicate with their constituents [21]. Similarly, only one qualitative study in rural Malawi investigated the perceived credibility of the sources of health information [24]. Participants in this study mentioned that they considered health information from health workers as the most credible, and that they would accept CCS only if recommended by them.

### 3.5. Correlation of Information Sources with CC Knowledge, Literacy, Screening Uptake and Attitudes toward CC and CCS

#### 3.5.1. CC Information Sources and CCS

Studies with rural samples only and those that oversampled participants from rural areas had CCS rates between 0.9% and 25.6% [3,15,26,29,35,38]. Studies that oversampled urban-based participants had uptake rates between 2.2% and 47.1% [25,31,32,39,40,42,43,46]. Rural women were statistically significantly less likely to report testing for CC compared to urban women (*p* = 0.005) in Namibia and Nigeria [13,32,39], and being an urban resident was a significant predictor of CCS [25,48] in Ethiopian and Burkinabe women. Among Namibian women, listening to the radio very often (odds ratio [OR] =1.17, *p* = 0.10) and watching television frequently (OR = 1.26, *p* =0.01) were associated with higher odds of CCS [32].

A lack of information/awareness of preventive screening was cited as the “main reason” for not screening in seven of eleven studies reporting reasons for or against screening [3,25,26,29,30,38,40,43]. Four of these eight studies had either rural-only or predominantly rural populations [3,26,29,38]. Seeking screening because of a healthcare provider’s recommendation was the most frequently stated reason among rural and urban women [11,22,33,37,43]. Initiation of CCS due to influence from community, mass media, or friends was reported by one study in Ethiopia [43].

#### 3.5.2. CC Information Sources and CC Knowledge

The proportion of women who had heard of CC ranged from 41 to 100%, while the proportion of women who had good knowledge of CC and CCS was between 15% and 56% [27,30,35,36,40,44,45]. Rural dwellers had poorer knowledge compared to urban residents [25,26,39,40,41,42]. For instance, among Nigerian women, rural women had significantly less knowledge about CC than urban or suburban subgroups (*p* = 0.005) [39]. Correspondingly, rural-based women in a sample from Ethiopia were 79% less likely to be knowledgeable about CC prevention measures compared to their urban counterparts (aAOR = 0.21, 95% CI: 0.18–0.34) [41].

There was a significant association between endorsing health professionals as sources of CC information with good CC knowledge among Ethiopian women aged 30–49 years (adjusted OR [AOR] = 2.3, 95% confidence interval [CI]: 1.27–4.17) [43]. Comparatively, a sample mainly composed of urban Ethiopian residents (56%) established a significant association between receiving information from healthcare providers and having adequate CC knowledge (AOR = 2.72, 95% CI: 1.69–4.37), compared to having mass media as the source of information [44]. In a predominantly rural sample of women in Ethiopia, those with access to any source of information regarding CC were 9.1 times (95% CI: 4.0–20.6) more likely to have good CC knowledge compared to those with none [15].

The use of television and radio as sources of information independently predicted good knowledge of CC among predominantly rural Ethiopian females aged 17–38 years (AOR = 1.918, 95% CI: 1.22–3.01) [35]. Comparably, a univariate analysis of a sample of rural and urban Ethiopian women aged 21 to 40 years, showed that having a functional radio or television was significantly associated with adequate CC knowledge (AOR = 2.72, 95% CI: 1.69–4.37) [45].

#### 3.5.3. CC Information and CC Literacy

Only one study conducted among literate rural Ugandan participants aged 18 to 65 years reported on the relationship between health information sources and CC literacy [13]. The study found that 97% of participants had limited CC literacy (print, oral, numeracy, and perceived e-health literacy), and there was a significant association between contact with health workers in health facility visits 12 months prior to the study and CC literacy (OR = 1.05, 95% CI: 1.02–1.07) [13]. However, CC literacy was not significantly associated with CCS (*p* = 0.204) in the study [13].

#### 3.5.4. Sources of CC Information and Screening Attitudes

The prevalence of negative attitudes toward CC and CCS was higher (59 to 95%) [11,15,26,33,36] compared to positive attitudes (46 to 85%) [11,30,34,35,36,37,40,41,43,48]. Negative attitudes toward CC and CCS were high in studies that sampled predominantly rural women [11,15,26,33,36]; one study in Tanzania specifically noted that fewer women from rural areas (63%) compared to those from urban areas (67%) thought that cervical cancer could be treated [26].

A good number of women were willing to screen (ranging between 38 and 90%) [13,26,27,33,34]. Others expressed willingness to screen only if screening was free and harmless (91%) [41], or only if they had CC symptoms [24]. Being less averse to screening was positively associated with a higher likelihood of seeking screening [37,47]. Some of the negative attitudes linked with CCS were being too young to contract the disease, the risk of stigmatization if screened, perceiving screening as a lack of faith in God, and fear of pain during screening, and fear of test results [27,39].

Only one study, with a sample in which 90% of the participants were rural-based, reported on the relationship between information sources and CC-related attitudes [15]. Receiving information from nurses compared to no information was associated with a 4.28 times higher likelihood (OR = 4.28, 95% CI: 2.4–7.4) of having a positive attitude towards CCS [15]. In the same study, receiving information from other sources (radio, family, newspaper, TV, poster, neighbors, or community) versus receiving none was associated with 5.06 times higher odds of having positive attitudes toward screening (OR = 5.06; CI: 2.48–10.33) [15].

## 4. Discussion

To the best of our knowledge, this is the first systematic review to assess the role of sources of information on CCS, CC knowledge, literacy, and attitudes toward screening among Sub-Saharan African women. Since our primary interest was rural populations that tend to have lower CCS uptake rates, we reviewed studies that sampled rural populations either in whole or partially. Studies have rarely reported on the preferences, credibility, and the role of CC information sources in shaping CC literacy and attitudes toward CC and CCS. However, understanding these factors is essential in the design of contextually relevant health communication strategies to increase the uptake of CCS in SSA [13,15,21,49].

Among women who had ever received any information about CC and CCS, health professionals and paraprofessionals were the most credible and most frequently reported sources [24]. The xisting literature from both high-income countries and low- and middle-income countries (LMICs) has echoed our findings, emphasizing the key role of healthcare workers as trusted sources of health information [50]. Although the contexts vary, women who receive information from healthcare providers may access accurate and more evidence-based information, and perhaps have opportunities to seek clarification when interacting with healthcare providers. Our findings suggest that expanding health providers’ efforts to community-wide CC information sharing is essential, particularly within rural areas in many LMICs that have limited public health information outlets and where women tend to present to hospitals with late-stage CC rather than for preventative screening [51,52].

The significance of mass media and print media in the dissemination of CC-related information among rural and urban women is evident in this review. A vast array of previous literature in other contexts supports the use of mass media in CC information dissemination. A study in Bangladesh, which used mass media for CCS campaigns found a 27% increase in Papanicolaou (Pap) testing uptake across all socio-economic groups [53]. Similarly, a time series analysis in the United Kingdom (UK) showed that mass media reporting of the experiences of a UK celebrity diagnosed with CC was associated with a significant increase in CCS among women aged 25 to 44 years [54]. Culturally relevant mass and print media, including the use of local dialects to enhance understanding among illiterate populations, is beneficial in CC prevention in SSA.

Social networks (including peers and relatives) served as another critical pathway for CC information, ranking third among the most frequently reported sources. Social networks are important avenues for health information sharing within LMICs. Studies have noted that health information disseminated through social networks can strongly influence not only an individual’s knowledge level but also subsequent decision-making for engaging in health prevention behaviors, such as CCS [10,55,56]. While our review did not delve into the specifics of the CC-related information shared nor the accuracy of such information, ensuring women’s ability to ascertain the accuracy of information shared within and across their social networks is paramount. 

Social media use in the acquisition of CC-related information was rare. As expected, two studies among rural women revealed that social media use was “common” among younger women [23] and less common among older women [11]. The two studies neither investigated the credibility of information shared on social media nor the association with CCS uptake, knowledge, literacy, and attitudes toward screening. Even so, the findings are critical given the heightened research on harnessing digital platforms for health information sharing within LMICs [11,57]. At the end of 2020, the Global System for Mobile Communications found that smartphone adoption was 64% in SSA, mostly among younger individuals, and was estimated to increase up to 75% by 2025 [58]. Most women diagnosed with CC in SSA are younger women aged 21 to 48 years [1]. However, prior research findings still indicate that compared to their urban counterparts, rural women are lagging in the adoption of smartphone, mainly due to limited access to electricity [59]. As governments in SSA implement strategies to eliminate CC, digital inclusion, encompassing population-based resource allocation to both rural and urban areas, will be vital in influencing smartphones use and the potential for disseminating CC information via social media.

Access to information sources, notably, healthcare workers, mass media, and social networks, was significantly linked to CCS uptake and good CC and CCS knowledge [15,32,43,44], as well as positive attitudes towards CC and CCS [15]. Women from rural areas had significantly lower CC knowledge [39,41] and more negative attitudes towards CC and CCS [11,15,26,33,36] than their urban counterparts, which may be attributed to limited access to information sources and CC preventive services in rural areas. Importantly, receiving CC information from health care providers was significantly associated with good CC and CCS knowledge compared to receiving information from mass media, further underscoring health care providers’ vital role in disseminating CC-related information. Of note, irrespective of the source, the CC information received seemed to promote positive attitudes toward CCS [15], indicating the beneficial impact of widespread CC information dissemination.

Our review found no evidence of a significant association between CC literacy and CCS uptake, but this finding was from only one study in rural Uganda [13]. This contradicts study findings among literate African immigrant women in the United States, which found a significant relationship between high CC literacy and Pap testing [60]. Perhaps African immigrant women in the study underwent screening because they had more confidence in receiving treatment for positive results, unlike the rural women in the Ugandan study, who may be faced with fears of the possibility of not accessing the necessary treatment if diagnosed with CC [61].

Finally, while analyzing CCS uptake and its relationship with information sources, we identified disparities. Rural and urban study populations had differences in access to certain information sources, with urban women having a wider range of CC information sources compared to rural women [22,26]. We also found disparities in CCS uptake, with studies overrepresenting rural populations reporting lower uptake compared to those overrepresenting urban populations. This discrepancy could stem from lack of access to screening and restricted access to information in rural areas, corroborated by multiple studies that identified a lack of information as the primary reason for not screening among rural women [3,26,29,30,38,40,43]. Governments and ministries of health in SSA should make intentional efforts to establish infrastructure including electricity in both rural and urban areas to support CCS information dissemination through multimedia platforms and the Internet [62]. Tailored interventions, designed with an understanding of specific challenges and needs of these populations, are crucial in bridging gaps.

### Limitations

While this review contributes valuable insights, it has limitations that should be considered when interpreting results. First, our findings are from only 13 countries in SSA, and therefore do not comprehensively represent CC information acquisition and use among all Sub-Saharan African women. Second, there are some methodological issues to be considered when interpreting this review’s findings. Although most of the included studies had high-quality ratings, the studies that sampled both rural and urban women had few proportions of rural women in the samples (2.3% to 45.6%) [25,30,31,32,39,40,42,43,44,45,46,48]. Some studies that investigated women’s screening practices failed to investigate the reasons for irregular CCS [13,15,23,27,31,32,36,37,42,46,47,48], which are vital considerations when designing interventions to enhance CCS. Except for one study [13], the studies that measured CC knowledge (a component of health literacy) [15,31,35,37,39,40,41,43,44,45], did not investigate other cancer-specific health literacy components including print, numeracy, communication, e-health, and information-seeking skills, which influence women’s capacity to seek CCS [16]. Additionally, although 64% of the included studies were conducted in community settings, rural and non-literate populations were underrepresented in most of the studies. Information-seeking behaviors and access to resources may differ greatly between urban literate women and rural non-literate women.

## 5. Conclusions

We found that Sub-Saharan African women had access to a variety of information sources that were positively associated with CC knowledge, literacy, screening uptake, and attitudes toward screening. There were disparities between CCS uptake, level of knowledge and attitudes toward CC and CCS among rural and urban populations. There is an urgent need for concerted efforts to improve CC information dissemination and boost CCS uptake, especially in resource-constrained rural regions. Among the studies included, there was limited research evidence on the credibility of and preferences for CC information sources, and the association of CC information sources with CC literacy and attitudes toward CCS.

## Figures and Tables

**Figure 1 ijerph-21-00872-f001:**
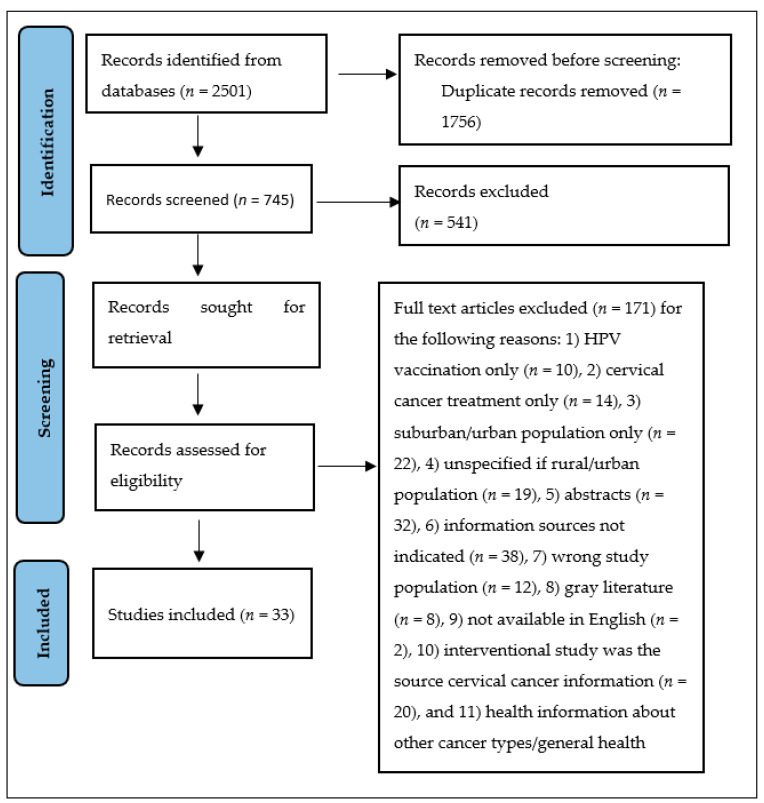
PRISMA flow diagram of screened and included articles.

**Table 1 ijerph-21-00872-t001:** Data extraction results.

Citation	Country	Study Participants	Study Setting	Study Design and Sampling	Cervical Cancer (CC) Information Sources and Preferences	Screening Practices, CC Knowledge, and Attitudes toward CC and Cervical Cancer Screening (CCS)	Association of Information Source with Cervical Cancer Screening Uptake, Knowledge, Literacy, and Screening Attitudes
Gafaranga et al., 2022 [3]	Rwanda	Women (N = 30) aged 30–59 years; other demographics unspecified	Rural = 16 (53.3%) and urban = 14 (46.7%)	Qualitative	Motivators for the use of screening services were personal (7), family members and friends (7), healthcare professionals (8), and government officials (SMS from telecommunication, radio, TV) (12).	Screening: Uptake not reported. Barriers to screening were pain/fear of a positive diagnosis (N = 26), high cost (N = 18), lack of information (N = 10), lack of insurance (N = 6), administrative barrier (absence of materials), and lack of accessible services (long distance to health facility (N = 3). Attitudes: Not studied Knowledge: Approx. 83% were aware of CC: 57% aware of prevention measures, and 77% had some knowledge/had heard of CCS.	Not applicable: qualitative study
Gatumo M et al., 2018 [11]	Kenya	Women (N = 451) ages 18–85 years selected using multi-stage cluster sampling: 14.2% were non-literate, 5.1% could read and write, though with no formal education, and 80.7% had a primary level of education or higher.	“Predominantly rural”	Descriptive cross-sectional	Approx. 79.8% (N = 360) of participants had heard about cervical cancer, and 15.1% (N = 301) had heard about HPV. Primary sources of information were family and friends (45.0%, n = 162), followed by a healthcare facility (40.3%, n = 145), radio/television (40.6%, n = 146), and less than 6.0% (n = 20) stated social media, newspaper, or a non-governmental organization.	Screening: Only 25.6% had ever undergone cervical cancer screening (N = 92). Attitudes: Approx. 89.2% of women who had heard about CC categorized it as “scary”, and 55.8% preferred a female health worker for cervical examination. About two-thirds perceived the examinations positively and believed that healthcare workers performing them were not rude to them. Knowledge: Approx. 80% were aware of CC; 44% scored above average on CC risk factors. Approx. 83.6% of women aware of CC had heard about CCS.	Not investigated
Jatho A et al., 2020 [13]	Uganda	Women (N = 400) aged 18–65 years: 53% (N = 212) with primary education level and 47% (N = 188) had a post-primary education level (non-literate participants not included).	Rural	Concurrent mixed methods	Radio, newspaper, mobile phones, health facilities	Screening: Only 24% (N = 96) of individuals had ever been screened for CC, but 99% of them were in the limited CC literacy category. Attitudes: Approx. 56% had intention to screen. Knowledge: Approx. 91% had had limited CC awareness.	Health facility visits during the previous 12 months (*p* = 0.025) and radio ownership (*p* = 0.024) were associated with cervical cancer literacy. Mobile phone ownership and newspaper reading frequency were not significantly associated with CC literacy. The relationship between information sources and CC screening was not investigated. CC screening was not significantly associated with CC literacy (*p* = 0.204).
Dozie et al., 2021 [14]	Nigeria	A random sample of women N (231) ages 15–40 attending antenatal clinic: 14.7% had no formal education, 26.4% had completed primary education, 43.7% had secondary education, and 15.2% had completed post-secondary education.	Rural area	Descriptive cross-sectional	The majority of participants had heard about CC screening (68.8%) from friends (52.85%), family/relatives (38.1%), health center/hospital (5.6%), the Internet (3%), and media (TV, radio, posters, newspapers) (0.4%). Few respondents had basic information on the cause of the disease (19%), prevention (13.9%), risk factors (20.8%), and treatment (23.4%). Approx. 80% of participants were aware of CCS locations.	Screening: Invasion of privacy (34.6%) and high cost of screening (29.4%) were strong reasons for avoiding screening. Attitudes: Only 37.6% were willing to screen. The majority (43%) reported they do not have money to waste on screening, would not get screened because of fear of the result (42%), and saw no need for screening (43.7%). Knowledge: Approx. 69% and 81% were aware of CCS and where to get services, respectively. The majority were not aware that CC is treatable.	Not investigated
Ruddies F et al., 2020 [15]	Ethiopia	Women (N = 341), mean age 35.5 years. The majority (63.5%; N = 217) had no formal education; 104—elementary school; 11—beyond 9 years of education; 9—tertiary education.	Rural = 307 (90.3%), urban = 34 (9.7%)	Descriptive cross-sectional	Approx. 50% of women had no source of information; nurses (32%), radio (9%), neighbors/community (5%), family (3%), newspaper/tv/poster (1%).	Screening: Only 2.3% (8) of women had been screened before, 70% had the intention to screen, and only 31% had access to a screening facility. Attitudes: Approx. 59% had a negative attitude toward CC and CCS, 13.5% considered themselves at risk for CC, and 61% thought it is a deadly disease. Knowledge: Approx. 36% (125) were aware of CC, 4.7% (14) knew symptoms, none knew HPV is a risk factor for CC. Approx. 88% were interested in learning more about CC.	Women with any source of information on CC and CCS were 9.1 times more likely to have good knowledge (CI:4.0–20.6) compared to those without any source. Naming nurses as a source of information was not significantly associated with good knowledge of CC and CCS. Women whose source of information was nurses were 4.28 times more likely to have a positive attitude towards CC, and those who named other sources had 5.06 odds of having a positive attitude towards CC. Women whose source of information was nurses had 21.05 times higher odds of CCS (OR = 21,0, CI:10.4–42.3), and indicating another source of information was associated with 5.8 odds of CCS (OR = 5.8, CI:2.4–13.5).
Adewumi et al., 2019 [21]	Kenya	The study included women who participated in an outreach and education campaign (N = 120), women screened (N = 111) using HPV testing via self-collected vaginal swabs, women who received treatment (N = 283), and women who were non-adherent to treatment (N = 72) and CHVs (N = 18).	Rural	Qualitative	Community leaders and village elders were considered essential for information dissemination on CC in the community due to their understanding of the functionality of their communities and the best ways to disseminate information to them. Few women mentioned gender preferences, but generally, community leaders are men in the Kenyan context. Male partners were also seen as sources of information on HPV screening and treatment.	Screening: Most women reported that they did not need their partners’ permission to seek screening. Attitudes: Stigmatizing attitudes toward HPV was linked to its association with promiscuity, HIV, and infidelity. Knowledge: Good CC knowledge may improve access to services and chances of getting male partner support for HPV-based CCS.	Not applicable: qualitative study
Nyambe N et al., 2018 [22]	Zambia	Women (N = 40; 19 HIV+, 19 HIV−, 2- unknown); men (N = 19) ages 25–49; education level data not collected	Rural and urban	Qualitative	The main source of CC information in both rural men and women in the study was health facilities. Other sources cited by the rural sample were family and friends, literature and media, community outreach activities work, and school (for women only). At the urban site, a health facility was the main source of information for women, but work/school was the major source of information for men. Other sources were family, friends and family, media and workplace, and community outreach activities in the urban sample. In contrast to rural residents, urban residents cited a wider range of information sources mainly from institutionalized sources (media, schools, and workplaces), which are more likely to be more accurate. Rural respondents were more likely to get information from friends, family, and community outreach activities.	Screening: Approx. 52.5% (N = 21/40) of women had been screened. Among HIV+ women, a lack of time (linked to long travel time to clinics and long waiting times) was a major barrier to screening, and in HIV- women, being symptomatic was mentioned as a determinant of CCS. Attitudes: Not investigated Knowledge: Women were generally knowledgeable about the role of CCS in CC prevention. HIV+ women had more accurate knowledge about risk factors and other prevention strategies compared to HIV- women. Rural residents were more likely to be aware of CC risk factors related to hygiene (e.g., douching) and traditional practices (e.g., circumcision) compared to urban residents.	Not applicable: qualitative study
Osei EA et al., 2021 [23]	Ghana	Females ages 19–60 years: N = 15 (42.9%) had secondary education, N = 13 (37.1%) had attained tertiary education, and N = 7 (20%) had basic education (elementary); non-literate participants not included.	Rural	Qualitative exploratory	Participants who had screened cited nurses and doctors as their sources of information on CC, followed by social media (WhatsApp, Facebook, and Instagram), followed by reproductive health education girls club and women fellowship meetings. Social media was second on the list since the sample was mainly composed of young women between the ages of 19 and 29 years. Other participants learned about CC and CCS from relatives, schools, television, and friends	Screening: “Few” participants had been screened. Attitudes: Not investigated Knowledge: The majority of the participants had little knowledge of the CCS types, and most of those who knew about the types of CCS were health professionals (nurses, pharmacists). Few knew about CCS costs and how screening is performed. Most were knowledgeable about CCS centers.	
Ports KA et al., 2015 [24]	Malawi	Women (N = 30) ages 18–46 years: education levels ranged from illiterate to secondary school completion (median of 6 years).	Rural	Qualitative	Nurses and doctors were mentioned by over 50% of the women. Few women mentioned written sources (newspapers, posters). Several mentioned village meetings. Doctors, nurses, and health surveillance assistants were reported as the most credible sources of health information. In terms of CCS and HPV services, women stated that they would accept them only if health professionals recommended them. Others mentioned that they would encourage the village headman to organize a group meeting so that other women in the community could learn about CC and CCS (desire to be community health advocates).	Screening: There was an uptake in the majority of women who had been screened because they were experiencing persistent vaginal bleeding. Lack of information was a key barrier to CCS. Attitudes: More than half of women whose perceived personal risk was low had never been screened and did not use protective measures during sexual intercourse. Some women who had not screened reported that they will only seek screening if they have symptoms (which are associated with late-stage CC), rather than preventive screening. Knowledge: Women universally expressed the need and desire for more information (about CC symptoms, screening, HPV and HPV vaccination).	Not applicable: qualitative study
Compaore S et al., 2016 [25]	Burkina Faso (N = 346), Togo (*n* = 5), Chad (N = 1)	Women (N = 351) aged 18–72 years: 25.4% (89) with no formal education, 21.6% (N = 76) with primary education, 35.6% (N = 125) with high school education level, and 17.4% (N = 61) with college/university-level education. Approx. 56.4% of rural residents in the sample were unemployed.	Urban = 294 (83.8%), rural = 39 (11.1%), and semi-urban = 18 (5.1%)	Descriptive cross-sectional	Health workers, relatives, friends	Screening: Approx. 54.7% of participants had never been screened. The main reason for not screening was a lack of awareness of CC and CCS, followed by not knowing where to get screened, fear of diagnosis with the disease, long distance to the hospital, and financial constraints. Urban residence (OR= 2.0; 95% CI: 1.19–3.25) and encouragement for screening by a healthcare worker (1.98; 95% CI: 1.06–3.69) were predictors of screening. Attitudes: Not reported Knowledge: Participants generally had a medium level of knowledge. There was higher knowledge among urban women (41.5%) compared to rural women (17.2%).	Women encouraged to have screening for medical reasons (advice from healthcare professionals or symptoms) were twice more likely to get screening (1.98; 95% CI: 1.06–3.69) than those with non-medical encouragement (relatives or friends’ advice).
Cunningham et al., 2015 [26]	Tanzania	Women (N = 575), ages 18–55 years. Education levels among rural residents: primary or less, 10.9% (33); completed secondary, 74.5% (225); college/university, 11.9% (36) and 1.7% (8). Education levels among urban residents: primary education or less, 5.5% (15); completed secondary, 56.1% (152); college/university, 32.5% (88) and 5.9% (16).	Rural =303 (52.7%) urban = 272 (47.3%)	Descriptive cross-sectional	The primary source of cervical cancer (CC) awareness was media (73%—radio; 22%—television; 13%—newspaper). Approx. 80% of rural women had heard about CC from the radio, followed by family/friend, church and healthcare providers, newspapers, and studies. Receiving information from church was more common among rural residents. The radio was also common among urban residents (64.3%), followed by television, family/friend, newspaper, studies, church, and healthcare. Only 13% of women had heard of CC from healthcare providers, and only 1% had heard from school. Churches, radio, and TV sources were significantly different between strata.	Screening: Approx. 4% rural and 9% urban had been screened. The largest barrier to screening was being unaware of the existence of preventive screening tests. Travel distance to healthcare facilities was a more frequently reported barrier in rural compared to urban women (27% versus 12%; *p* < 0.001). About 50% anticipated that the cost of screening or the travel costs would be prohibitive, and 25% stated that the opportunity cost of taking a leave of absence from work would be a barrier. Attitudes: Approx. 90% acceptability of screening among women who had never screened; more rural women were willing to travel more than 2 h to access screening (60% vs. 49%, *p* < 0.001). Rural women were more likely than urban women to believe that CC cannot be treated (63.1% versus 67.2%). Knowledge: Approx. 13% (*n* = 62) of the sample—17.9% (*n* = 40) of urban and 8.8% (*n* = 22) of rural—had adequate CC knowledge.	Not investigated
De Kubber et al., 2011 [27]	South Africa	Women (N = 532) ages 35–49 years; no other demographics provided	Rural	Descriptive cross-sectional	The main source of information was word of mouth from nurses in clinics (40%), followed by friends/relatives (44%); only 16% received information from the media (radio and television).	Screening: Uptake not studied. Attitudes: Approx. 90% were willing to screen. Among those unwilling, 23% and 25% mentioned fear of pain and test results, respectively. Knowledge: Approx. 60% and 74% of participants had heard about CC and Pap smear, respectively. Less than 50% knew the signs of CC.	Not investigated
Endalew DA et al., 2020 [28]	Ethiopia	Women of reproductive age (N = 268) ages 15–49 years: 18.5% were illiterate, 34 had primary education level (13%), 67 (25.8%), had secondary education, and 111 (42.7%) had college or above.	Rural and urban	Descriptive cross-sectional	Approx. 56.0% of respondents acquired information about cervical cancer screening from mass media, while 25.2% received information from family, friends, and neighbors; 11% from brochures and printed materials; and 7.8% from health workers.	Screening: Only 3.8% of respondents had ever been screened for cervical cancer. The majority of participants (N = 133 (53.2%) endorsed a lack of health education programs to promote cervical cancer screening as the main barrier to screening, and 11.6% stated long distance to screening facility. Approx. 30% thought that that screening would be too expensive, and 5.2% were afraid of a positive diagnosis for CC. Attitudes: Not studied Knowledge: Approx. 83.8% had heard about CC; 77% were unaware of its symptoms; between 0.4% and 9% knew about risk factors; and 98% did not know about CCS.	Women who had information about cervical cancer were 10 times more likely to have been screened for cervical cancer [(AOR = 10.2 (95%CI1.9–96.4)].
Ezem BU, 2007 [29]	Nigeria	Women aged 20–65 years (N = 846). Approx. 630 (74.5%) had tertiary education, 124 (14.7%) had secondary education, and 24 did not indicate education level.	Rural town	Descriptive cross-sectional	Out of the 447 respondents who were aware of screening (52.8%), 140 received information from hospital sources, 138 (30.9%) got their information from friends, 94 (21%) from books or magazines, 40 from school, 13 from their husbands, 10 from television/radio, and 13 from other sources (e.g., those who had had cervical cancer).	Screening: only 7.1% (60) had ever been screened. The most common reasons for not screening were lack of awareness 390 (46.1%), did not see a need for it 106 (12.5%), fear of a bad result Approx. 98 (11.6%) did not know where screening can be performed; were not recommended by their doctor, 46 (5.4%); screening was too expensive, 46 (5.4%); and the rest endorsed other reasons or did not state 63 (7.4%). Attitudes: Not studied Knowledge: Approx. 52.8% of respondents were aware of CC.	Not investigated
Gebisa et al., 2022 [30]	Ethiopia	Women (N = 414) aged 18–49 years	Urban = 322 (77.8%), and rural = 92 (22.2%)	Descriptive cross-sectional	Media (N = 149), health workers (N = 38), family/friends/neighbor (N = 60), religious leaders (N = 10), teachers (N = 20), written materials (N = 8)	Screening: Approx. 6.3% of women had been screened. Among those who had not been screened for cervical cancer, the most commonly reported reasons for not screening for cervical cancer were lack of information about the procedure (N = 187) and thinking they were healthy (N = 123). Attitudes: Approx. 46.1% had positive attitudes towards CCS; 26% agreed that precancerous CCS methods could be helpful in CC prevention. Knowledge: Approx. 69% had heard about CC, and 50.7% had good CC knowledge.	Not investigated
Isabirye A et al., 2020 [31]	Uganda	Women (N = 845) ages 25–49 years: only 57.6% had attained at least secondary education, 42.4% had attained some primary education, and 13.0% had no formal education.	Urban = 600 (71%), rural = 245 (29%)	Descriptive cross-sectional	Cervical cancer screening information: radio (15%), health worker (42%), TV (16%), others (10%). Cervical cancer screening was higher among women whose main source of information about cervical cancer screening was health workers (41.5%).	Screening: Approx. 21% had been screened in their lifetime. Attitudes: Not investigated Knowledge: Approx. 58% had high CC and CCS knowledge. Women with high knowledge of CC and CCS were more likely to have screened (26.2%).	In univariate analysis, cervical cancer screening was significantly associated with the source of information (*p* < 0.001).
Kangmennaang J et al., 2015 [32]	Namibia	This analysis was focused on a sub-sample of women (N = 6542) aged 15–64 years who had heard about cervical cancer from a national sample; 357 (5.46%) had no formal education, 1198 (18.31%) had a primary education, 4301 (65.74%) had a secondary education level, and 686 10.49%) had tertiary education or higher.	Urban = 3756 (57.41%) Rural = 2786 (42.59%)	Descriptive cross-sectional	Healthcare providers in health facilities, radio, television.	Screening: Only 38.92% (N = 2546) had ever been screened. Compared to urban residents, rural residents (OR = 0.68, *p* = 0.01) were less likely to report testing for cervical cancer. Attitudes: Not investigated Knowledge: Approx. 65.81% had heard of CC.	Having contact with health personnel in the last 12 months (OR = 1.35, *p* = 0.01) was also significantly associated with being screened for cervical cancer. Those who listen to the radio very often (OR =1.17, *p* = 0.10) and watched television frequently (OR = 1.26, *p* = 0.01) were more likely to report screening for cervical cancer.
Kasa et al., 2018 [33]	Ethiopia	Women (N = 735) ages 17–88 years: 111 (15.1%) were illiterate, and 248 (33.7%) had a diploma in their educational status	“Predominantly rural”	Descriptive cross-sectional	Sources of information about cervical cancer were family and friends (51.5%); health workers (19.8%); others (7.9%); media (16.5%) (type of media unspecified); brochures, posters, and other printed materials (4.3%).	Screening: Approx. 7.3% had been screened in their lifetime, most of whom had screened only once (83%); 68.5% of those screened endorsed self-initiation, while 31.5% were offered by health professionals. Reasons for not screening were lack of symptoms (45.4%) (N = 334), fear of the result (21.8%), not being informed (20.1%) (N = 160), and perception that that screening would be painful (5.3%). Attitudes: The majority had a negative attitude (63%) towards CCS; 48% were willing to screen; 36% thought that screening was not expensive; and 41% and 70% agreed that screening causes no harm, and any adult woman can acquire CC. Knowledge: Approx. 69.3% (N = 509) of participants had heard about cervical cancer; 142 (19.3%) had heard about CC screening.	Not investigated
Kimondo FC et al., 2021 [34]	Tanzania	Women (N = 297) ages 18–55: 4% (N = 12) had no formal education, 73.1% primary level education (N = 217), and 22.9% (N = 68) had secondary level education or more. Approx. 10.8% had health insurance.	Rural and urban regions	Descriptive cross-sectional	The major source of cervical cancer screening information was healthcare providers (80.1%), radio (42.8%), TV (23.2%), awareness campaigns (17.2%), relative/friends (9.1%), and print media (8.1%). Approx. 94.3% (*n* = 280) endorsed ownership of information technology devices (devices not specified in the study).	Screening: Approx. 50.2% (N = 149) had been screened for CC, and 64.4% (N = 96) of them were screened in the previous 12 months. Reasons for screening in the previous 12 months were mainly due to provider advice 88.6% (N = 132), followed by HIV status (22.8%), screening campaigns (18.8%), age (1.3%), and support from partners (0.7%). Among those unscreened, reasons for not screening were no symptoms (53.4%), uninformed about screening location (25%), unaware/busy (22.3%), delays in obtaining service (14.9%), fear of pain (11.5%), fear of results (8.8%), and expense (8.1%). Attitudes: The majority regarded screening as necessary (89%) and were willing to screen (79%). Overall, 67% had a positive attitude toward screening, and 71.4% were comfortable with screening by a healthcare provider of any gender. Knowledge: The majority had heard about CCS (90%) but only 21% knew the timing of CCS for women living with HIV. The majority were aware of CC signs and symptoms (71%) and ways to prevent it (53%).	Not investigated
Mruts KB & Gebremariam TB, 2018 [35]	Ethiopia	Female undergraduate students (N = 571) ages 17–38 years; illiterate participants not included.	Rural = 357; (62.5%) and urban = 214 (37.5%)	Descriptive cross-sectional	Approx. 232 participants (40.5%) had heard about cervical cancer. The main sources of information were mass media (TV, radio), 134 (58%) and health institutions, 44 (19.0%).	Screening: Only 5 (0.9%) had ever been screened. The main reasons for not screening were lack of information, 276 (55.6%) and fear of being infected in 96 (19.4%) of participants. Attitudes: Approx. 33% considered themselves at risk for CC, 85% considered CC a severe disease, 65% and 75% believed its curable and preventable through screening, respectively. Knowledge: Approx. 41% had heard about CC, and 36% generally had good knowledge.	Using radio and TV as sources of information [AOR = 1.918 (95% CI: 1.223, 3.010)] and having information about STIs [AOR = 3.030 (95% CI: 1.665, 5.514) were independent predictors of good knowledge of CC.
Mukama T et al., 2017 [36]	Uganda	Women (N = 900) aged 25–49 years: 142 (15.8%) had no formal education, 530 (58.9%) completed primary, and 228 (25.3%) completed secondary education.	Rural = 610 (67.8%), semi-urban = 195 (21.7%), urban = 195 (10.5%)	Descriptive cross-sectional	Radio was the main source of information about cervical cancer (N = 557; 70.2%), followed by health centers (N = 129; 15.1%), and friends/family members (N = 104; 13.1%).	Screening: Practices not reported Attitudes: The majority considered CC a severe disease (95%), curable if diagnosed early (78%), and symptomatic (83%). Most considered themselves at high risk for CC (76%) and believed that screening was important (94.4%). Knowledge: Approx. 99.8% (N = 898) of women had heard about CC, and 88.2% (N = 794) had heard about CC screening; 55% had high knowledge of CC and its risk factors.	Not investigated
Mwantake et al., 2022 [37]	Tanzania	Women living with HIV (WLHIV) (N = 297) aged 18–55 years: 229 (77.1%) had primary education or less, 68 (22.9%) had secondary education or above. Approx. 10.8% had health insurance. Illiterate individuals not included.	Rural districts and urban region	Descriptive cross-sectional	Healthcare providers (HCPs) (N = 238; 80.1%) and mass media (N = 145; 48.8%). Approx. 94.3% (N = 280) owned any form of technology.	Screening: Approx. 50.2% of the WLHIV had been screened for CC. Attitudes: Approx. 66.7% had a positive attitude towards CCS, and those who had a positive attitude towards screening (59%) had higher odds of CCS uptake compared to those with a negative attitude. Knowledge: About 72% had inadequate knowledge of CC signs and symptoms.	Women whose source of cervical cancer screening information was HCPs were 17.3 times more likely to have screened for CC compared to those who received information from other sources.
Ndejjo R et al., 2016 [38]	Uganda	Females (N = 900) aged 25–49 years: 672 (74.7%) had no/primary education, and 228 (25.3%) had post-primary education	Rural = 610 (67.8%) and semi-urban = 290 (32.2%)	Descriptive cross-sectional	Health workers (48.8%); other sources not investigated	Screening: Only 43 (4.8%) (N = 20 rural residents and N = 23 urban/semi-urban residents) had ever been screened; 21 were screened because a health worker had requested, 16 self-initiated to know their status, and 17 did so after experiencing signs and symptoms associated with cervical cancer. Among participants who had not been screened, 416 (48.5%) stated that they were not aware, and 142 indicated health facility-related challenges (distance, costs, waiting times) and personal reasons (lack of time, fear of test outcomes, not being at risk, lack of time). Attitudes: Not investigated Knowledge: Knowing at least one method of CCS and someone who had ever been screened or diagnosed were positively associated with CCS uptake.	Recommendation for CC screening and knowing where CC screening was provided were independent predictors of CC screening ((*p* = 0.01) and (*p* = 0.04), respectively). Women who had been recommended for CC screening were 87 times more likely to have been screened.
Rimande-Joel R and Ekenedo GO, 2019 [39]	Nigeria	Women (N = 978) of childbearing age (15–49 years); no formal education (15.3%; N = 150), primary (N = 184), secondary (N = 369), tertiary (N = 275)	Urban = 440 (45%), rural = 254 (26%), semi-urban = 284 (29%)	Descriptive cross-sectional	Nurses—during antenatal health education in government hospitals, health talks by political leaders (on diseases that affect women)	Screening: Approx. 45.2%, 25.6%, and 28.3% of women regularly, occasionally, or had never engaged in screening/prevention practices, respectively. Rural women had the poorest CCS and prevention practices. Attitudes: Albeit possessing appropriate knowledge, women generally had negative attitudes toward CC and its preventive measures (e.g., thought they were too young to contract CC, risk stigmatization if screened, believed screening is painful and showed a lack of faith in God). Knowledge: Rural residents had the least knowledge about CC.	Not investigated
Tadesse A et al., 2022 [40]	Ethiopia	Graduate female students who were registered for the academic year 2013/2014; 15–20 years (N = 568), >20 years (N = 99); year 1 (38.7%), year 2 (32.1%), year 4 (18.7%), and year 4 and above (10.5%); non-literate participants not included	Rural = 170 (25.5%) and urban = 497 (74.5%)	Descriptive cross-sectional	Among women who heard about CC (60.6%), the most common source of information was news media (57.4%), and the least was religious leaders (1.7%). The second source was health workers (32.9%), followed by family/neighbor/friend (19.3%), teacher (17.1%), and brochure/other (16.6%). The majority of participants had poor knowledge scores (85.2%).	Screening: Only 2.2% had screened for CC in their lifetime. Lack of information was the most reported reason for not screening (42% (283), followed by lack of symptoms (28%), not yet decided (15%), feeling shy (5.5%), and high cost of screening (2.5%). Attitudes: Approx. 72% agreed that CC is fatal, 66% perceived that any woman can acquire CC, and 73% agreed that CCS helps in prevention. Knowledge: Approx. 61% had heard about CC, but only 15% had good CC knowledge. Those born in urban areas were twice more knowledgeable than those born in rural areas.	Not investigated
Tafere Y et al., 2021 [41]	Ethiopia	Women (N = 844) ≥18: 59.1% with formal education and 40.9% with no formal education	Rural = 514 (60.1%) and urban = 330 (39.1%)	Descriptive cross-sectional	Approx. 66% of respondents had heard about CC mainly from health professionals (75.4%), and 10.6% used the media (radio, TV), and other sources not mentioned.	Screening: Uptake not reported. Lack of awareness and unfavorable attitudes towards screening impede CC prevention. Attitudes: Approx. 64% of respondents had a favorable attitude towards CC prevention measures, and 91% would agree to be screened if the test is free and creates no harm. Approx. 50% believed to be at risk, 48% agreed that CC is prevalent in Ethiopia, 62% agreed that it is non-communicable, 71% agreed that screening helps prevent CC, 65% perceived screening causes no harm, and 60% thought screening for CC is expensive. Knowledge: Approx. 75% had poor knowledge. Being a rural resident was negatively associated with knowledge of CC prevention measures (AOR = 0.21, 95%CI; 0.18, 0.34).	Not investigated
Tapera et al. 2019 [42]	Zimbabwe	Women (N = 277; N = 143 community sample and N = 134 hospital sample), aged ≥25: 71% had completed secondary education, 17% higher education, 13% primary; there were no illiterate participants in the sample.	Rural = 110 (39.7%) and urban = 167 (60.3%)	Descriptive cross-sectional	In the community sample (143), among those who had ever screened (N = 42), sources of CC information were radio (*n* = 30; 73%), TV (N = 8; 20%), health workers (0), and other (e.g., churches) (N = 3; 7%)	Screening: Approx. 29% (42) of healthy women had been screened; 86% of these were from urban areas, and 14% were from rural areas. The majority of those who had never been screened were <45 years old. More than half (52%) of women who had ever been screened were affiliated with Protestant and Pentecostal churches. Healthy women who had never visited doctors/health facilities in the last 6 months were less likely to screen. Attitudes: Not investigated Knowledge: The majority (74%) knew that CC is treatable.	Source of information was not associated with CC screening services after controlling for confounders (radio: OR = 1.18 (CI:0.02–58.95)); TV: OR = 5.48, CI (0.08–396).
Tekle T et al., 2020 [43]	Ethiopia	Women (N = 516) aged 30–49 years: 21.3% (*n* = 110) had no formal education, 20% (N = 103) had some primary school education, 24.4% (*n* = 126) had attended secondary school, and 34.3% (N = 177) had a diploma/degree.	Rural = 171 (33.1%) and urban = 345 (66.9%)	Descriptive cross-sectional	Healthcare workers (52.5%), community, friends, mass media	Screening: Approx. 77.1% (398) had never screened for CC. Of those screened, 22.9% (118) had been screened only once; 52.5% were screened at the initiation of a health professional; (43.2%) were self-initiated; and the rest were initiated by community, mass media, or friends. Lack of information was the main reason for not screening and was mentioned by 83.7% of respondents. Other reasons were lack of a screening service (69.8%), long wait time (51.4%), lack of respect by health professionals (26%), expensive service cost (11.8%), and negative perception of procedure (10.5%). Only 15.2% (N = 26) of rural residents (N = 171) had screened, compared to 26.7% (*n* = 92) of urban residents (N = 345). Urban residents were 2 times more likely to have screened compared to rural residents 2.03 (1.25, 3.28). Attitudes: Approx. 46% had a favorable attitude toward CC and CCS. Knowledge: Approx. 43% had good CC and CCS knowledge.	Having sourced information about CC from a health professional was associated with good knowledge of CCS (AOR = 2.3, 95% CI: 1.27–4.17)); knowing someone who had CC was associated with screening for CC at least once (AOR = 2.47, 95% (1.37–4.44)).
Wakwoya EB et al., 2020 [44]	Ethiopia	Women aged 25–49 years (N = 1181): 33.7% were illiterate, 32.3% had primary education, 19.8% had completed some level of secondary education, and 14.1% had completed high school or more.	Urban = 821 (69.5%) and rural N = 360 (30.5%)	Descriptive cross-sectional	Mass media was the main source of information, followed by health professionals and friends/neighbors in both categories of participants who had inadequate knowledge and those with adequate knowledge.	Screening: Uptake not reported Attitudes: Not investigated Knowledge: Approx. 49% had heard about CC, and 56% had adequate knowledge.	Having adequate cervical cancer knowledge was significantly associated with receiving information from healthcare providers (AOR: 2.72, 95% CI 1.69–4.37) compared to having mass media as the source of information.
Woldu BF et al., 2020 [45]	Ethiopia	Women (N = 237) ages 21–40 years: 32.1% with no formal education, 16.5% with primary education, 24.1% with secondary education, and 27.4% with a diploma or above	Rural = 105 (44.3%), urban, = 132 (55.7%)	Descriptive cross-sectional	Health professionals were the predominant sources of information (66.1%). Mass media 38.6% (*n* = 51) Printed materials 3.8% (N = 5) Health professionals 64.4% (N = 85) Family/friends 4.8% (N = 6)	Screening: Practice not investigated Attitude: Not investigated Knowledge: Only 56% had heard about CC; 52% had good knowledge of CC and CCS. Approx. 84.6% of women knew where screening services were offered. Urban residence compared to rural residence was associated with good knowledge.	Having a functional TV/radio was associated with adequate knowledge of cervical cancer in a bivariate analysis (*p* < 0.2) but not in a multivariate analysis.
Tiiti et al., 2022 [46]	South Africa	Women (*n* = 526) ≥18 years (mean age was 36.8 years); educational level not included	Rural = 12 (2.3%), semi-urban = 447 (85%), urban N = 63 (12%)	Analytic cross-sectional study	Approx. 7% (N = 37) of those who had screened (47.1%) reported that the screening had been recommended by a health worker.	Screening: Approx. 47.1% of the participants had been previously screened for CC using Papanicolaou (Pap) test. Attitudes: Not investigated Knowledge: Approx. 40%, 32%, 21%, and 6.3% had no knowledge, fair, good, and very good knowledge of HPV, respectively. Those previously screened had a higher likelihood of very good knowledge of HPV (60.6%). Only 19% currently mentioned causes/risk factors for CC.	Not investigated
Perng P et al., 2013 [47]	Tanzania	Women (N = 300) aged 25–49 years. Approx. 202 women were recruited from home, and 98 women were recruited during a 2-day free screening intervention. Approx. 4% had no formal education, 5% had some primary education, 74% completed primary or higher, and 22% were missing data.	Rural	Analytic cross-sectional	Radio; over 75% of the sample owned a radio.	Screened: Approx. 35% were screened during the study period. Attitudes: Those who were least averse to screening (23%) compared to the most averse (4.3%) were more likely to attend a screening service. Knowledge: CC risk factors (34%) and screening knowledge (18%) were positively associated with attending CCS.	Women who attended screening listened to the radio regularly (OR 24.76; 95% CI, 11.49–53.33, listened to the radio 1–3 times per week versus not at all) and were older (OR 4.29; 95% CI, 1.61–11.48, age 40–49 years versus 20–29 years) compared to women who did not.
Ayanto et al., 2022 [48]	Ethiopia	Women (N = 410 ages 30–49 years). Women who were eligible for screening and had been screened within the last 5 years and 28 of the control group (eligible for screening but not yet screened in the last 5 years) (case vs. control). No formal education N = 23 (11.2%), N = 27 (13.2%); primary education (1–8) N = 97 (47.1%), N = 112 (54.9%), secondary education (9–12) N = 47 (22.8%), N = 44 (21.6%), tertiary education (12+) N = 39 (18.9%), N = 21 (10.3%)	Rural = 287 (45.6%), urban =223 (54.4%)	Unmatched case-control	Among women who were informed about cervical cancer and CCS (N = 67.8%), sources of information were (case vs. control) health workers, 149 (73%) and 28 (13.6%); mass media, 43 (21.1%) and 39 (18.9%); social network, 8 (3.9) and 5 (2.4%); and Women’s Development Army, 3 (1.5%).	Screening: Urban residence was a significant predictor of screening utilization [AOR: 2.7, 95% CI: (1.56, 4.56)]. Attitudes: Approx. 48.5% of women who received CCS during the study period and within 5 years (cases) had favorable attitude towards CC and CCS compared to 47% (*n* = 96) of controls. Knowledge: Approx. 62% of the cases and 97% of controls had good CC knowledge.	Not investigated

Approx.—approximately.

**Table 2 ijerph-21-00872-t002:** Quality Appraisal.

First Author and Year	Study Design	Methodological Quality Criteria	Appraisal (Based on Reviewer 1 and 2 Consensus)	
			Yes, No, Cannot Tell	Comments
Gafaranga et al., 2022 [3]	Qualitative study	1.1. Is the qualitative approach appropriate to answer the research question? 1.2. Are the qualitative data collection methods adequate to address the research question? 1.3. Are the findings adequately derived from the data? 1.4. Is the interpretation of the results sufficiently substantiated by the data? 1.5. Is there coherence between qualitative data sources, collection, analysis and interpretation?	1.1. Yes 1.2. Yes 1.3. Yes 1.4. Yes 1.5. Yes	N/A
Gatumo M et al., 2018 [11]	Descriptive cross-sectional study	4.1. Is the sampling strategy relevant to address the research question? 4.2. Is the sample representative of the target population? 4.3. Are the measurements appropriate? 4.4. Is the risk of nonresponse bias low? 4.5. Is the statistical analysis appropriate to answer the research question?	4.1 Yes 4.2 Yes 4.3 Yes 4.4 Yes 4.5 Yes	N/A
Jatho A et al., 2020 [13]	Concurrent mixed-methods design	5.1. Is there an adequate rationale for using a mixed-methods design to address the research question? 5.2. Are the different components of the study effectively integrated to answer the research question? 5.3. Are the outputs of the integration of qualitative and quantitative components adequately interpreted? 5.4. Are divergences and inconsistencies between quantitative and qualitative results adequately addressed? 5.5. Do the different components of the study adhere to the quality criteria of each tradition of the methods involved?	5.1: Yes 5.2: Yes 5.3 Yes 5.4 No 5.5 Yes	Divergence and inconsistencies between qualitative and quantitative results were not reported.
Dozie et al., 2021 [14]	Descriptive cross-sectional study	4.1. Is the sampling strategy relevant to address the research question? 4.2. Is the sample representative of the target population? 4.3. Are the measurements appropriate? 4.4. Is the risk of nonresponse bias low? 4.5. Is the statistical analysis appropriate to answer the research question?	4.1: Yes 4.2. Yes 4.3. Yes 4.4. Yes 4.5. Yes	N/A
Ruddies F et al., 2020 [15]	Descriptive cross-sectional study	4.1. Is the sampling strategy relevant to address the research question? 4.2. Is the sample representative of the target population? 4.3. Are the measurements appropriate? 4.4. Is the risk of nonresponse bias low? 4.5. Is the statistical analysis appropriate to answer the research question?	4.1 Yes 4.2 Yes 4.3 Yes 4.4 Yes 4.5 Yes	N/A
Adewumi et al., 2019 [21]	Qualitative study	1.1. Is the qualitative approach appropriate to answer the research question? 1.2. Are the qualitative data collection methods adequate to address the research question? 1.3. Are the findings adequately derived from the data? 1.4. Is the interpretation of results sufficiently substantiated by data? 1.5. Is there coherence between qualitative data sources, collection, analysis, and interpretation?	1.1. Yes 1.2. Yes 1.3 Yes 1.4 Yes 1.5 Yes	N/A
Nyambe N et al., 2018 [22]	Qualitative study	1.1. Is the qualitative approach appropriate to answer the research question? 1.2. Are the qualitative data collection methods adequate to address the research question? 1.3. Are the findings adequately derived from the data? 1.4. Is the interpretation of results sufficiently substantiated by the data? 1.5. Is there coherence between the qualitative data sources, collection, analysis, and interpretation?	1.1. Yes 1.2. Yes 1.3. Yes 1.4. Yes 1.5. Yes	N/A
Osei EA et al., 2021 [23]	Qualitative exploratory design	1.1. Is the qualitative approach appropriate to answer the research question? 1.2. Are the qualitative data collection methods adequate to address the research question? 1.3. Are the findings adequately derived from the data? 1.4. Is the interpretation of results sufficiently substantiated by the data? 1.5. Is there coherence between the qualitative data sources, collection, analysis, and interpretation?	1.1 Yes 1.2 Yes 1.3 Yes 1.4 Yes 1.5 Yes	N/A
Ports KA et al., 2015 [24]	Qualitative study	1.1. Is the qualitative approach appropriate to answer the research question? 1.2. Are the qualitative data collection methods adequate to address the research question? 1.3. Are the findings adequately derived from the data? 1.4. Is the interpretation of the results sufficiently substantiated by the data? 1.5. Is there coherence between the qualitative data sources, collection, analysis, and interpretation?	1.1 Yes 1.2 Yes 1.3 Yes 1.4 Yes 1.5 Yes	N/A
Compaore 2016 [25]	Descriptive cross-sectional study	4.1. Is the sampling strategy relevant to address the research question? 4.2. Is the sample representative of the target population? 4.3. Are the measurements appropriate? 4.4. Is the risk of nonresponse bias low? 4.5. Is the statistical analysis appropriate to answer the research question?	4.1: Yes 4.2: Yes 4.3 Yes 4.4 Yes 4.5 Yes	N/A
Cunningham et al., 2015 [26]	Descriptive cross-sectional study	4.1. Is the sampling strategy relevant to address the research question? 4.2. Is the sample representative of the target population? 4.3.Are the measurements appropriate? 4.4. Is the risk of nonresponse bias low? 4.5. Is the statistical analysis appropriate to answer the research question?	4.1: Yes 4.2: Yes 4.3 Yes 4.4 Yes 4.5 Yes	N/A
De Kubber et al., 2011 [27]	Descriptive cross-sectional study	4.1. Is the sampling strategy relevant to address the research question? 4.2. Is the sample representative of the target population? 4.3. Are the measurements appropriate? 4.4. Is the risk of nonresponse bias low? 4.5. Is the statistical analysis appropriate to answer the research question?	4.1 Cannot tell 4.2 Cannot 4.3 Yes 4.4 Yes 4.5 Yes	The sampling strategy was unspecified; exclusion and inclusion criteria were unclear.
Endalew DA et al., 2020 [28]	Descriptive cross-sectional study	4.1. Is the sampling strategy relevant to address the research question? 4.2. Is the sample representative of the target population? 4.3. Are the measurements appropriate? 4.4. Is the risk of nonresponse bias low? 4.5. Is the statistical analysis appropriate to answer the research question?	4.1 Yes 4.2 Yes 4.3 Yes 4.4 Yes 4.5 Yes	N/A
Ezem BU, 2007 [29]	Descriptive cross-sectional study	4.1. Is the sampling strategy relevant to address the research question? 4.2. Is the sample representative of the target population? 4.3. Are the measurements appropriate? 4.4. Is the risk of nonresponse bias low? 4.5. Is the statistical analysis appropriate to answer the research question?	4.1 Yes 4.2 Yes 4.3 Yes 4.4 Cannot tell 4.5 Yes	4.4 Cannot tell if nonresponse bias is low since the number of surveys distributed was not reported.
Gebisa et al., 2022 [30]	Descriptive cross-sectional study	4.1. Is the sampling strategy relevant to address the research question? 4.2. Is the sample representative of the target population? 4.3. Are the measurements appropriate? 4.4. Is the risk of nonresponse bias low? 4.5. Is the statistical analysis appropriate to answer the research question?	4.1: Yes 4.2: Yes 4.3 Yes 4.4 Yes 4.5 Yes	N/A
Isabirye A et al., 2020 [31]	Descriptive cross-sectional study	4.1. Is the sampling strategy relevant to address the research question? 4.2. Is the sample representative of the target population? 4.3. Are the measurements appropriate? 4.4. Is the risk of nonresponse bias low? 4.5. Is the statistical analysis appropriate to answer the research question?	4.1 Yes 4.2 Yes 4.3 Yes 4.4 Yes 4.5 Yes	N/A
Kangmennaang J et al., 2015 [32]	Descriptive cross-sectional study	4.1. Is the sampling strategy relevant to address the research question? 4.2. Is the sample representative of the target population? 4.3. Are the measurements appropriate? 4.4. Is the risk of nonresponse bias low? 4.5. Is the statistical analysis appropriate to answer the research question?	4.1 Yes 4.2 Yes 4.3 Yes 4.4 Yes 4.5 Yes	N/A
Kasa et al. 2018 [33]	Descriptive cross-sectional study	4.1. Is the sampling strategy relevant to address the research question? 4.2. Is the sample representative of the target population? 4.3. Are the measurements appropriate? 4.4. Is the risk of nonresponse bias low? 4.5. Is the statistical analysis appropriate to answer the research question?	4.1 Yes 4.2 Yes 4.3 Yes 4.4 Yes 4.5 Yes	N/A
Kimondo FC et al., 2021 [34]	Descriptive cross-sectional study	4.1. Is the sampling strategy relevant to address the research question? 4.2. Is the sample representative of the target population? 4.3. Are the measurements appropriate? 4.4. Is the risk of nonresponse bias low? 4.5. Is the statistical analysis appropriate to answer the research question?	4.1: Yes 4.2: Yes 4.3 No 4.4 Yes 4.5 Yes	The questionnaire was not pretested, and there was no mention of its validation processes and reliability tests.
Mruts KB and Ge-bremariam TB, 2018 [35]	Descriptive cross-sectional study	4.1. Is the sampling strategy relevant to address the research question? 4.2. Is the sample representative of the target population? 4.3. Are the measurements appropriate? 4.4. Is the risk of nonresponse bias low? 4.5. Is the statistical analysis appropriate to answer the research question?	4.1 Yes 4.2 Yes 4.3 Yes 4.4 Yes 4.5 Yes	N/A
Mukama T et al., 2017 [36]	Descriptive cross-sectional study	4.1. Is the sampling strategy relevant to address the research question? 4.2. Is the sample representative of the target population? 4.3. Are the measurements appropriate? 4.4. Is the risk of nonresponse bias low? 4.5. Is the statistical analysis appropriate to answer the research question?	4.1: Yes 4.2: Yes 4.3 Yes 4.4 Yes 4.5 Yes	N/A
Mwantake et al., 2022 [37]	Descriptive cross-sectional study	4.1. Is the sampling strategy relevant to address the research question? 4.2. Is the sample representative of the target population? 4.3. Are the measurements appropriate? 4.4. Is the risk of nonresponse bias low? 4.5. Is the statistical analysis appropriate to answer the research question?	4.1 Yes 4.2 Yes 4.3 Yes 4.4 Yes 4.5 Yes	N/A
Ndejjo R et al., 2016 [38]	Descriptive cross-sectional study	4.1. Is the sampling strategy relevant to address the research question? 4.2. Is the sample representative of the target population? 4.3. Are the measurements appropriate? 4.4. Is the risk of nonresponse bias low? 4.5. Is the statistical analysis appropriate to answer the research question?	4.1: Yes 4.2: Yes 4.3 Yes 4.4 Yes 4.5 Yes	N/A
Rimande-Joel R and Ekenedo GO, 2019 [39]	Descriptive cross-sectional study	4.1. Is the sampling strategy relevant to address the research question? 4.2. Is the sample representative of the target population? 4.3. Are the measurements appropriate? 4.4. Is the risk of nonresponse bias low? 4.5. Is the statistical analysis appropriate to answer the research question?	4.1 Yes 4.2 Yes 4.3 Yes 4.4 Yes 4.5 Yes	N/A
Tadesse A et al., 2022 [40]	Descriptive cross-sectional study	4.1. Is the sampling strategy relevant to address the research question? 4.2. Is the sample representative of the target population? 4.3. Are the measurements appropriate? 4.4. Is the risk of nonresponse bias low? 4.5. Is the statistical analysis appropriate to answer the research question?	4.1 Yes 4.2 Yes 4.3 Yes 4.4 Yes 4.5 Yes	N/A
Tafere Y et al., 2021 [41]	Descriptive cross-sectional study	4.1. Is the sampling strategy relevant to address the research question? 4.2. Is the sample representative of the target population? 4.3. Are the measurements appropriate? 4.4. Is the risk of nonresponse bias low? 4.5. Is the statistical analysis appropriate to answer the research question?	4.1 Yes 4.2 Yes 4.3 Yes 4.4 Yes 4.5 Yes	N/A
Tapera et al. 2019 [42]	Descriptive cross-sectional study	4.1. Is the sampling strategy relevant to address the research question? 4.2. Is the sample representative of the target population? 4.3. Are the measurements appropriate? 4.4. Is the risk of nonresponse bias low? 4.5. Is the statistical analysis appropriate to answer the research question?	4.1 Yes 4.2 Yes 4.3 Yes 4.4 Yes 4.5 Yes	N/A
Tekle T et al., 2020 [43]	Descriptive cross-sectional study	4.1. Is the sampling strategy relevant to address the research question? 4.2. Is the sample representative of the target population? 4.3. Are the measurements appropriate? 4.4. Is the risk of nonresponse bias low? 4.5. Is the statistical analysis appropriate to answer the research question?	4.1 Yes 4.2 Yes 4.3 No 4.4 Yes 4.5 Yes	4.3 The survey was adapted from a previous study but not pretested; no reports are available on its reliability in this study and the study that originally used the survey.
Wakwoya EB et al., 2020 [44]	Descriptive cross-sectional study	4.1. Is the sampling strategy relevant to address the research question? 4.2. Is the sample representative of the target population? 4.3. Are the measurements appropriate? 4.4. Is the risk of nonresponse bias low? 4.5. Is the statistical analysis appropriate to answer the research question?	4.1 Yes 4.2 Yes 4.3 Yes 4.4 Yes 4.5 Yes	N/A
Woldu BF et al., 2020 [45]	Descriptive cross-sectional study	4.1. Is the sampling strategy relevant to address the research question? 4.2. Is the sample representative of the target population? 4.3. Are the measurements appropriate? 4.4. Is the risk of nonresponse bias low? 4.5. Is the statistical analysis appropriate to answer the research question?	4.1 Yes 4.2 Yes 4.3 Yes 4.4 Yes 4.5 Yes	N/A
Tiiti et al., 2022 [46]	Analytic cross-sectional study	3.1. Are the participants representative of the target population? 3.2. Are the measurements appropriate regarding both the outcome and intervention (or exposure)? 3.3. Are there complete outcome data? 3.4. Are the confounders accounted for in the design and analysis? 3.5. During the study period, is the intervention administered (or exposure occurred) as intended?	3.1 Yes 3.2 Yes 3.3 Yes 3.4 Yes 3.5 Yes	N/A
Perng P et al., 2013 [47]	Analytic cross-sectional study	3.1. Are the participants representative of the target population? 3.2. Are the measurements appropriate regarding both the outcome and intervention (or exposure)? 3.3. Are there complete outcome data? 3.4. Are the confounders accounted for in the design and analysis? 3.5. During the study period, is the intervention administered (or exposure occurred) as intended?	3.1 Yes 3.2 Yes 3.3 Yes 3.4 Yes 3.5 Yes	N/A
Ayanto et al., 2022 [48]	Unmatched case-control	3.1. Are the participants representative of the target population? 3.2. Are the measurements appropriate regarding both the outcome and intervention (or exposure)? 3.3. Are there complete outcomes of the data? 3.4. Are the confounders accounted for in the design and analysis? 3.5. During the study period, is the intervention administered (or exposure occurred) as intended?	3.1 Yes 3.2 Yes 3.3 Yes 3.4 Yes 3.5 Yes	N/A

## Data Availability

Data are contained within the article.

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
