# Peer review of "The Role of Health Information Sources on Cervical Cancer Literacy, Knowledge, Attitudes and Screening Practices in Sub-Saharan African Women: A Systematic Review"

_ijerph, 2024, doi:10.3390/ijerph21070872_

Round 1
Reviewer 1 Report
Comments and Suggestions for Authors
This is a very well-written systematic review exploring on the role of health information sources in influencing cervical cancer screening.
Here are some comments to consider:
1. It would be better to define health literacy in the introduction to reflect its meaning in the context of cervical cancer screening.
2. For the title, I feel that focusing on the role of health information sources on the different outcome variables related to cervical cancer screening would reflect more the purpose of the study.
3. For the future directions section, it would be interesting if this section were to be embedded in the discussion and more elaborated on based on the lessons learned and findings from the study.
Author Response
- It would be better to define health literacy in the introduction to reflect its meaning in the context of cervical cancer screening.
Thank you for this comment. We revised the sentence to:
“CC screening literacy was operationally defined as the degree to which women can obtain, process, understand, and communicate information related to cervical cancer screening; while CC knowledge was defined as the vocabulary and conceptual understanding of CC information.” [page 2, line 71 to 74]
- For the title, I feel that focusing on the role of health information sources on the different outcome variables related to cervical cancer screening would reflect more the purpose of the study.
We agree. The title has now been revised to:
“Reducing Geographical Disparities in Cervical Cancer Prevention in Sub-Saharan Africa: A Systematic Review of Health Information Sources” (P.1 line 3-4)
- For the future directions section, it would be interesting if this section were to be embedded in the discussion and more elaborated on based on the lessons learned and findings from the study.
Thank you for this suggestion. We embedded the paragraph in future direction in the discussion section based on findings and lessons learned from the study.
In place of the paragraph in future directions section: “.... Studies rarely reported on the preferences, credibility, and the role of CC information sources in shaping CC literacy and attitudes toward screening. However, understanding these factors are in the design of contextually relevant health communication strategies to increase the uptake of CCS in SSA [13,15,21,49].” [page 49, lines 291-295]
“ ....Tailored interventions, designed with an understanding of specific challenges and needs of these populations, are crucial in bridging gaps.” (page 50, line 373-374]
Reviewer 2 Report
Comments and Suggestions for Authors
Dear Respectable Authors
Thank you for considering a great area of research related to cervical cancer. You conducted a systematic review to identify cervical cancer information sources and their correlation with cervical cancer- -screening uptake, knowledge, literacy, and attitudes toward screening among Sub-Saharan African women. Your results are of interest but the way you report the manuscript needs some revisions as follows;
- Lines 20 and 69, there is a punctuation. Please add a "," after Embase.
- Abstract, please add search period or exact date of search.
- Abstract, please add some information regarding eligibility criteria, quality appraisal checklist, and data extraction.
- Please add statistical data to the method as you mentioned a correlation between variables. How do you estimate this correlation (Line 26)?
- Abstract, conclusion, you must state what was the level of knowledge, attitude, and practice among this population.
- The introduction section is short. Please add more information regarding knowledge, attitude, and practice and also mention the gap in this field.
- It is good to update your search but why January 2023?
- Lines 91-4 are not related to this section. these statements are related to results under the "study selection" heading. See PRISMA statement 2020.
- Also, remove Figure 2 to results. In line 92, you stated Figure 2 but in the text, there is only one Figure. Please refine it.
- Line 98, this table is one of your main results in a systematic review and must mentioned under the subheading "characteristics of the included studies" in the results section. Please check the PRISMA statement for details.
- Line 108-10 are related to results under the subheading "quality appraisal or assessment.
- Please explain the reasons for heterogeneity.
- Results, please separate section 3.1 into two subheadings based on PRISMA 2020. Also, add Figure 1 and Table 1 at the end of these two sections.
- Please add more results regarding quality assessment in the results section. Please prepare results based on items on the MMAT checklist.
Cheers
Comments on the Quality of English LanguageThere are several punctuation in the text. Please read it again.
Author Response
Thank you for considering a great area of research related to cervical cancer. You conducted a systematic review to identify cervical cancer information sources and their correlation with cervical cancer- -screening uptake, knowledge, literacy, and attitudes toward screening among Sub-Saharan African women. Your results are of interest but the way you report the manuscript needs some revisions as follows;
- Lines 20 and 69, there is a punctuation. Please add a "," after Embase.
Thank you for the suggestion. We added a comma in lines 20 and 69 (Page 2 , line 79 )
- Abstract, please add search period or exact date of search.
Thank you for the suggestion. The search period and exact date has been added.
“Peer-reviewed literature was searched on March 2, 2022, and updated on January 24, 2023, in four databases.” (Page 1, lines 18-19 )
- Abstract, please add some information regarding eligibility criteria, quality appraisal checklist, and data extraction.
Thank you for the suggestion. We have added this information as follows:
eligibility criteria: “Eligible studies were empirical, published after 2002, included rural women, and reported on women’s information sources and preferences.” (Page 1, lines 19-20)
quality appraisal checklist: “The quality of selected articles was assessed using the Mixed Methods Appraisal Tool.” (Page 1, lines 20-21)
data extraction: “Data extraction was conducted on an Excel Spreadsheet...” (Page 1, lines 21-22)
- Please add statistical data to the method as you mentioned a correlation between variables. How do you estimate this correlation (Line 26)?
Thank you for the comment. We recognize that the use of the term ‘correlation’ may have implied that we performed statistical analysis. We did not conduct a correlation analysis but performed a content analysis. The sentence has been edited as follows: “A content analysis indicated a positive association of information sources with cervical cancer literacy, knowledge, screening, and positive screening attitudes.” (Page 1, lines 29-31)
- Abstract, conclusion, you must state what was the level of knowledge, attitude, and practice among this population.
Thank you for the suggestion. The level of cervical cancer knowledge, attitude, and practice has been updated: “There was generally low cervical cancer knowledge, literacy, and screening uptake, yet high prevalence of negative attitudes toward cervical cancer and its screening; these outcomes were worse in rural areas” (Page 1, lines 26-28)
- The introduction section is short. Please add more information regarding knowledge, attitude, and practice and also mention the gap in this field.
We agree with this comment and suggestion. The introduction section has now been udpated with additional information regarding knowledge, attitude and practice and the gap in the field mentioned:
Screening practice: “Only about 14% of women in SSA between ages 30 and 49 have ever been screened for CC in their lifetime and approximately 12% have been screened at least twice by age 45, which is lower than the 70% target stipulated by the World Health Organization [2].” (Page 1-2, lines 40-43)
Knowledge: “....Particularly, basic information on the causes of CC, its risk factors and signs and symptoms, the benefits of screening in early detection of CC, and location of screening services, is sparsely disseminated to women in SS Africa [13].” (Page 2, lines 54-57)
Attitudes: “Some of the negative attitudes that tend to hinder CCS include fear of the screening procedure and fatalistic beliefs about CC [14]. (Page 2, lines 57-58)
Gap: “Nonetheless, the role of information sources in shaping women’s CC knowledge, literacy and attitudes toward screening and screening uptake is rarely explored. Research is also limited in characterizing information sources considered acceptable and credible in the context of cervical cancer prevention among women in SSA. To design interventions aimed at preventing CC and reducing CC burden attributable to limited access to reliable and accurate CC information [13], it is essential to understand the role of health information sources in CC knowledge, literacy, attitudes toward screening and screening uptake among Sub-Saharan African women, especially in rural areas where screening uptake rates are very low.
(Page 2, lines 58-66)
- It is good to update your search but why January 2023?
Thank you for the question. We updated the search to determine if there was any newly published literature relevant to the study purpose. The sentence has not been updated to: “A literature search was conducted on March 2nd, 2022, and updated on January 24th, 2023, to determine whether there was any include newly published evidence on the research topic.” (Page 2, lines 77-78)
- Lines 91-4 are not related to this section. These statements are related to results under the "study selection" heading. See PRISMA statement 2020.
Thank you for the suggestion. The statements and the figure have been moved to the results section.
“Figure 1 below shows the Preferred Reporting Items for Systematic Reviews and Meta-analysis (PRISMA) flow diagram of records identified, screened, and included [17].” (Page 3, lines 125-127)
- Also, remove Figure 2 to results. In line 92, you stated Figure 2 but in the text, there is only one Figure. Please refine it.
Thank you for noticing this discrepancy. Figure 2 has been edited to “Figure 1”. (Page 3, lines 125-127)
- Line 98, this table is one of your main results in a systematic review and must mentioned under the subheading "characteristics of the included studies" in the results section. Please check the PRISMA statement for details.
We agree. Table 1 is now mentioned under ‘Features of included studies’. “Table 1 shows a summary of the results.” (Page 4, line 131)
- Line 108-10 are related to results under the subheading "quality appraisal or assessment.
We agree. Lines 108-110 “Inter-rater agreement (IRA) ranged from 40% to 100% (an average IRA of 85%) with 96% (n=32) of these rated as high quality.” belonged to the results section. However, we deleted this sentence as it did not align with the reporting guidelines on the tool used (MMAT). The updated quality appraisal is as follows:
“A summary of quality assessment of included literacy is shown on Table 2 below. Most studies met all five methodological quality criteria used to assess each one, based on the study design [3,11,14,15,21–26,28,30–33,35–42,44–48]. Only one study failed to meet two of five quality assessment criteria due to failure to specify the sampling strategy and ambiguity in its exclusion and inclusion criteria [27]. Therefore, reviewers were unable to determine whether the sampling strategy was relevant in addressing the research question and if the sample was representative. One study did not report the number of surveys distributed and it was impossible to establish the existence of non-response bias [29]. One mixed methods study failed to report on the divergences and inconsistencies between qualitative and quantitative results [13]. Two studies neither pre-tested adapted instruments nor reported their reliability, hence reviewers could not establish the appropriateness of the measures in addressing the research questions [34,43].” (Page 34, lines 161-173)
- Please explain the reasons for heterogeneity.
Thank you for the question. Studies included in this review had variability in clinical outcomes (cervical cancer- knowledge, attitudes, screening, and literacy), and study designs (qualitative, analytic and descriptive cross-sectional designs and unmatched case control). This sentence has now been updated to:
“Due to clinical and methodological heterogeneity, a narrative synthesis, as opposed to a meta-analysis, was used to report the findings of this review [19]. (Page 3, lines 117-118)
- Results, please separate section 3.1 into two subheadings based on PRISMA 2020. Also, add Figure 1 and Table 1 at the end of these two sections.
Thank you for the suggestion. Section 3.1 has been separated into “Study selection” and “Features of included studies”. Figure 1 (page 4, lines 128) and Table 1 (page 5, lines 157) have been added at the end of these sections
- Please add more results regarding quality assessment in the results section. Please prepare results based on items on the MMAT checklist.
Thank you for the suggestion. More information regarding quality assessment has been added and results prepared in a table format based on MMAT checklist (Table 2) (page 34-46 lines 135-177)
“Most studies met all five methodological quality criteria used to assess each, based on the study design [2,10,13,14,20–25,27,29–32,34–41,43–47]. Only one study failed to meet two of five quality assessment criteria due to failure to specify the sampling strategy and ambiguity in its exclusion and inclusion criteria [26]. Therefore, reviewers were unable to determine whether the sampling strategy was relevant in addressing the research question and if the sample was representative. One study did not report the number of surveys distributed and it was impossible to establish the existence of non-response bias [28]. One mixed methods study failed to report on the divergences and inconsistencies between qualitative and quantitative results hence reviewers [12]. Two studies neither pre-tested adapted instruments nor reported their reliability [33,42].” (Page 34, lines 161-173)
Round 2
Reviewer 2 Report
Comments and Suggestions for Authors
Dear Respectable Authors
Thank you for your clarification. Your article has improved a lot, but a few issues have not been followed, which are described below.
- Why did you remove the names of databases from the abstract? Please add the databases here.
- The title of the manuscript still does not match the purpose of the study. What is the reason for that? Please choose a title that fits the purpose (s) of the study.
- About a year and a half has passed since your last search. Why are the databases checked in the new update only until the beginning of 2023?
- The first item in the method section of a systematic review is eligibility criteria. Please correct it. Pay attention to item 5 of the checklist.
Cheers
Author Response
Thank you for your clarification. Your article has improved a lot, but a few issues have not been followed, which are described below.
- Why did you remove the names of databases from the abstract? Please add the databases here.
Thank you for the question. We were restricted by the word limit on the abstract section. The names of databases have been added (Page 1, lines 20-21)
- The title of the manuscript still does not match the purpose of the study. What is the reason for that? Please choose a title that fits the purpose (s) of the study.
Thank you for the comment: The title was changed in response to a comment from the first reviewer (“For the title, I feel that focusing on the role of health information sources on the different outcome variables related to cervical cancer screening would reflect more the purpose of the study”). However, we realize that we may have misunderstood the comment.
The title has been updated to: ‘The Role of Health Information Sources on Cervical Cancer Literacy, Knowledge, Attitudes and Screening Practices in Sub-Saharan African Women: A Systematic Review’
- About a year and a half has passed since your last search. Why are the databases checked in the new update only until the beginning of 2023?
Thank you for your question. The databases were last updated in January 2023, before we finalized our analysis and drafted the manuscript. Since then, it has taken considerable time to thoroughly analyze the findings and complete the manuscript. Additionally, we submitted this manuscript to a journal in October 2023, which took about 4 months to get feedback and eventually resulted in the manuscript being rejected.
We updated the search today (June 26, 2024) and identified 136 articles that may qualify for inclusion. We request additional time (about 2 months) to screen the abstracts and full texts of these articles to ensure the manuscript is as comprehensive and current as possible.
- The first item in the method section of a systematic review is eligibility criteria. Please correct it. Pay attention to item 5 of the checklist.
Thank you for the comment. The eligibility criteria section has been moved and is now the first item on the checklist (page 2, lines 76-85)